
# Dominance of climate warming effects on recent drying trends over wet monsoon regions

Chang-Eui Park[1,2], Su-Jong Jeong[1], Chang-Hoi Ho[2], Hoonyoung Park[2], Shilong Piao[3], Jinwon Kim[4], Song Feng[5]

[1]School of Environmental Science and Engineering, South University of Science and Technology of China, Shenzhen, 518055, China
[2]School of Earth and Environmental Sciences, Seoul National University, Seoul, 08826, South Korea
[3]College of Urban and Environmental Sciences, Peking University, Beijing, 100871, China
[4]Department of Atmospheric and Oceanic Sciences, University of California, Los Angeles, 90024, CA, USA
[5]Department of Geosciences, University of Arkansas, Fayetteville, 72701, AR, USA

Jan 2017
Atmospheric Chemistry and Physics

*Correspondence to*: Su-Jong Jeong (waterbell@gmail.com)

**Abstract**

Understanding changes in continental surface dryness is key information for adapting to climate change because of the critical socioeconomic consequences. Recent studies reveal that spatial patterns of continental dryness trends are in contrast to the "dry gets drier, wet gets wetter" paradigm. Causes of the complexity in dryness trends remain uncertain because various climate parameters control continental dryness. Here, we quantify the relative effects of dominant climate drivers on dryness trends over continental East Asia, which is characterized by diverse hydro-climate regimes ranging from humid to arid, by analyzing observed data from 189 weather stations for the period of 1961-2010. Since the early 1980s, monsoon climate zones (east of 100°E) have been getting significantly drier, but the related mechanisms vary according to the hydro-climate regime. Drying trends in arid regions are mostly explained by reduced precipitation. In contrast, in humid areas, the increase in evapotranspiration due to increased atmospheric water-holding capacity, a secondary impact of warming, is the primary condition for the increase in dryness. This drying impact of atmospheric moisture deficiency is much stronger in humid areas than in arid areas. Our results suggest that enhanced atmospheric water demands caused by warming can threaten water resources in wet monsoon areas and possibly in other warm and water-sufficient regions.





## 1 Introduction

The mechanism behind changes in continental fundamentally differs from that over the ocean because of limited surface

moisture availability (Hoekstra and Mekonnen, 2012; Greve et al., 2014; Sherwood and Fu, 2014; Hegerl et al., 2015). In many

assessments, precipitation ($P$), the amount of water supply, is regarded as a key variable for understanding variations in dryness,

particularly in humid regions such as Asian monsoon regions (Wang et al., 2012; Kitoh et al., 2013; Liu and Allan, 2013). For

example, in East Asia, dryness changes are generally summarized as "the dry western region (west of 100°E) is getting wetter,

the dry northern region (east of 100°E and north of 35°N) is getting drier, and the wet southeastern region (east of 100°E and

south of 35°N) is getting wetter" based on changes in annual mean $P$ (Wang and Ding, 2006; Piao et al., 2010). In addition, a

decrease in $P$ leads to drying trends over the northern and central-east regions of India, part of the South Asian monsoon region

(Zhou et al., 2008; Roxy et al., 2015). However, climate change significantly varies potential evapotranspiration ($PET$) (Liu

et al., 2010; Han et al., 2012; Shan et al., 2012), the amount of atmospheric moisture demand. $PET$ variations largely affect

dryness trends that are in turn closely related to the occurrence of droughts, water scarcity, and tree mortality (Westerling et

al., 2006; Williams et al., 2013; Dai, 2013). Drying impacts of $PET$ increase are usually emphasized in water-limited regions

(Westerling et al., 2006; Estes et al., 2014); however, humid areas are also expected to experience severe aridification in the

21st century because of a continuous increase in $PET$ (Feng and Fu, 2013; Cook et al., 2014). Thus, the processes involved in

the variability of dryness need to be examined over various hydro-climate regimes to better understand continental dryness

changes.

This study aims to elucidate the mechanisms of dryness trends in continental East Asia through the analysis of observed climate

data at 179 and 10 weather stations in mainland China and South Korea, respectively, for the period 1961−2010. The long-

term trend in dryness is a critical concern for continental East Asia, as it is a region of massive populations, widely varying

hydro-climate regimes, fragile ecosystems, and significant agricultural activities (Piao et al., 2010; Geng et al., 2014; Jeong et

al., 2014). Also, the analysis region has recently experienced abrupt climate changes (Gong and Ho 2002; Yue et al., 2013).

For example, northeast China experienced severe warming by 0.36 ºC decade⁻¹ for the period of 1960-2006 (Piao et al. 2010).

Rainfall intensity has significantly increased over southeastern China (Zhai et al., 2005). Previous assessments of trends in

surface dryness show contradictory results over continental East Asia. Assessments based on grid reanalysis data generally





suggest that continental East Asia is getting drier due to an increase in *PET* accompanied by an increase in the vapor pressure
deficit (VPD) (Feng and Fu, 2013; Greve et al., 2014; Huang et al., 2016). On the contrary, the other studies using site
observations reported that more than half of the stations over mainland China show negative trends in both *PET/P* and *PET*,
indicating a decrease in surface dryness, following a decrease in solar irradiance and wind speed despite continuous warming
(Wu et al., 2006; Zhang et al., 2009; Huang et al., 2016). Thus, a quantitative analysis is needed to explain the contradiction
between previous assessments regarding surface dryness over continental East Asia.
In this study, an aridity index, *PET/P*, defined as *PET* based on the Penman–Monteith equation (Penman, 1948; Allen et al.,
1998) divided by *P*, is employed to assess surface dryness and its trends (Middleton et al., 1997; Estes et al., 2014; Greve et
al., 2014). Over land, the amount of actual evaporation (*AET*) is constrained by the amount of *P*, which is also generally less
than *PET* because of limited available water at the land surface (Fu and Feng, 2014; Greve et al., 2014). Thus, the *PET/P* ratio
is more suitable for measuring the degree of water deficiency or surplus for a certain climate condition. If the value of *PET/P*
is less than unity, the location is classified as a wet region, and vice versa. Likewise, as the aridity index decreases, the land
surface becomes wetter, and vice versa. By the definition of the aridity index, trends in surface dryness can be resolved by
combining the effects of changes in five climate parameters: *P*, net radiation (*Rn*), wind speed (*WS*), surface air temperature
(*Ta*), and relative humidity (*RH*). Furthermore, we classify the analysis domain into three hydro-climate regimes based on the
50-year climatology of *PET/P*: arid (*PET/P* ≥ 2), transitional (1 ≤ *PET/P* < 2), and humid (*PET/P* < 1) (Geng et al., 2014) (Fig.
S1). The ratio *PET/P* and regional classification allow the identification of climate parameters that are important for trends in
surface dryness over the three hydro-climate regimes.

## 2 Methods and data

### 2.1 Climate dataset

Climate data for the period 1961−2010 are obtained from 179 and 10 meteorological sites in mainland China and South Korea,
respectively. Data include daily mean air temperature, precipitation, wind speed at a height of 10 m, relative humidity, and
sunshine duration. The quality of this data is controlled by the National Meteorological Center of the China Meteorological





Administration and Korea Meteorological Administration. The meteorological sites satisfy the following criteria: 1) the
existence of all climate parameters in the year 2010, 2) sufficient records for at least 10 years for the two analysis periods (i.e.,
1961–1983 and 1984–2010).
**2.2 Calculation of daily PET**
Daily *PET* values are calculated from the Penman-Monteith approach, which is one of the credible methods for estimating
atmospheric water demand (Sheffield et al., 2012). The formulation of daily *PET* following the Penman-Monteith approach is
written as:
$$PET = \frac{\Delta}{\Delta + \gamma} R_n + \frac{\gamma}{\Delta + \gamma} \frac{c_1 (1 + c_2 U_2)(e_s - e_a)}{\lambda} \qquad (1)$$

where $\Delta$ is the slope of the vapor pressure curve (kPa K$^{-1}$) at a certain temperature, $\gamma$ is the psychrometric constant (kPa K$^{-1}$),
$R_n$ is the net radiation at the surface (mm day$^{-1}$), $c_1$ is 6.43 MJ kPa$^{-1}$ day$^{-1}$, $c_2$ is 0.536 s m$^{-1}$, $U_2$ is the wind speed at a height
of 2 m (m s$^{-1}$), $e_s$ is the saturation vapor pressure of the air (kPa), $e_a$ is the actual vapor pressure (kPa), and $\lambda$ is the latent heat
of vaporization (MJ mm$^{-1}$) (Allen et al., 1998; Sheffield et al., 2012). This *PET* equation is a simplified form of the FAO
Penman-Monteith equation that neglects  stomatal conductance and heat flux from the ground. All of the variables are
computed using the station-based climate data following an equation set that is described in the FAO56 report (Allen et al.,
1998). The wind speed at a height of 2 m is computed from station-observed wind speed at 10 m using a wind profile
relationship (Han et al., 2012). Station elevations are computed by linear interpolation and Global 30 Arc-Second Elevation
(GTOPO30) of the United States Geological Survey to estimate the net radiation based on sunshine duration. There are
differences between the interpolated elevation and actual elevation due to the limitation of spatial resolution, but the temporal
variation of *PET* or the relative influence of climate parameters cannot be changed with the elevation differences.
**2.3 Change-point analysis**
We use two methods to find the change-point of the temporal variation of *PET/P*. One method defines the change-point when
cumulative sum of *PET/P* variation for the $i$th year ($C_i$) is greatest (Pettitt, 1980). The cumulative sum $C_i$ is provided as follows:
$$C_0 = 0 \qquad (2)$$

$$C_i = C_{i-1} + (X_i - \bar{X}) \qquad (3)$$



where $X_i$ is the *PET/P* anomaly in year $i$, and $\bar{X}$ is the averaged *PET/P* for the whole analysis period. In the other change-point
model (Elsner et al., 2000), $X_i$ is the same, *PET/P* of the $i$th year. $Y_i$ is defined as $\log_{10}(X_i + 1)$. The step variable $T_i$ is defined
for an integer $p$ that changes from 2 to $q = N - 1$ as follows:
$$T_i(p) = \begin{cases} 0, & i < p \\ 1, & i \geq p \end{cases} \quad (4)$$

where $N$ is the total number of years of the analysis period 1961–2010. Using the step-variable $T_i$, a simple linear first-order
regression model is suggested for an integer $p$ as follows:
$$Y_i = \alpha_0(p) + \alpha_1(p)T_i(p) + \epsilon_i(p) \quad (5)$$

where $\alpha_0(p)$ is the intercept, $\alpha_1(p)$ the slope and $\epsilon_i(p)$ the error of residual at $Y_i$ for a fixed $p$. In addition, the value of $P(p)$
is computed by
$$P(p) = \hat{\alpha}_1(p)/se[\hat{\alpha}_1(p)] \quad (6)$$

where $se[\hat{\alpha}_1(p)]$ is the standard error of $\alpha_1(p)$. Let $P(p_1) = max\{|P(2)|, \ |P(3)|, \ ... \ , \ |P(q)|\}$. The $p_1$ can be a change-point
if the $P(p_1)$ is statistically significant.
**2.4 Estimation of the relative influences of climate parameters**
The derivative of the aridity index with respect to time is written using the following equation:
$$\frac{d}{dt}\left(\frac{PET}{P}\right) = -\frac{PET}{P^2}\frac{dP}{dt} + \frac{1}{P}\frac{dPET}{dt} \quad (7)$$

The first and second terms on right-hand side indicate temporal changes in the aridity index due to changes in $P$ and *PET*. PET
can be decomposed into four climate parameters using multilinear regression:
$$PET = a_{Rn}R_n + a_{WS}WS + a_{Ta}T_a + a_{RH}RH + b \quad (8)$$

where $a_{Rn}, a_{WS}, a_{Ta},$ and $a_{RH}$ are the regression coefficients of *Rn*, *WS*, *Ta*, and *RH*, respectively, and the constant $b$ is the
intercept. We obtain the time derivative of Eq. (8) as follows:
$$\frac{dPET}{dt} = a_{Rn}\frac{dR_n}{dt} + a_{WS}\frac{dWS}{dt} + a_{Ta}\frac{dT_a}{dt} + a_{RH}\frac{dRH}{dt} \quad (9).$$

where each term on the right-hand side indicates trends in *PET* with respect to changes in each climate variable individually.
Finally, Eq. (7) is written as follows:




$$\frac{d}{dt}\left(\frac{PET}{P}\right) = -\frac{PET}{P^2}\frac{dP}{dt} + \frac{1}{P}\left(a_{Rn}\frac{dR_n}{dt} + a_{WS}\frac{dWS}{dt} + a_{Ta}\frac{dT_a}{dt} + a_{RH}\frac{dRH}{dt}\right)$$

$$\approx -\frac{\overline{PET}}{\overline{P}^2}\frac{dP}{dt} + \frac{1}{\overline{P}}\left(a_{Rn}\frac{dR_n}{dt} + a_{WS}\frac{dWS}{dt} + a_{Ta}\frac{dT_a}{dt} + a_{RH}\frac{dRH}{dt}\right) \qquad (10)$$

where the terms on the right-hand side indicate the trend in the aridity index considering changes in $P$, $Rn$, $WS$, $Ta$, and $RH$,
sequentially. $\bar{P}$ and $\overline{PET}$ are the average of $P$ and $PET$ for the analysis period, respectively.

## 3 Results

### 3.1 Changes in dryness trends over continental East Asia during 1961-2010

Figure 1 depicts temporal variations in mean $PET/P$, $P$, and $PET$ for all stations expressed as annual mean anomalies. For the
entire period, $PET/P$ decreases at a rate of -2.30% decade[-1] due to both increases in $P$ (2.44% decade[-1]) and decreases in $PET$
(−0.52% decade[-1]), implying reduced dryness caused by increased water supply as well as decreased atmospheric water
demands. However, the temporal variation in $PET/P$ is not monotonic. The change-point of the long-term trend in $PET/P$ is
1983 based on two change-point analyses. This change-point is significant at the 99% confidence level. The trend in $PET/P$ is
negative (-1.81% decade[-1]) for 1961−1983 and positive (1.66% decade[-1]) for 1984−2010 (Fig. 1a). The decrease in $PET/P$
before the early 1980s is due mainly to the relatively large increase in $P$ (4.56% decade[-1]) rather than the decrease in $PET$ (-
0.95% decade[-1]) (Figs. 1a and 1b). In contrast, the increase in $PET$ (1.22% decade[-1]) largely contributes to the increase in
$PET/P$ during the later period (Figs. 1a and 1c).
The spatial distributions of $PET/P$, $P$, and $PET$ trends are consistent with those of the overall changes in both periods (Fig. 2).
Note that the scale of $P$ trends (Figs. 2b and 2e) is reversed in order to represent drying and wetting trends as red and blue
colors, respectively. For the earlier period, 60% of the total number of stations show decreasing trends in $PET/P$, particularly
in the arid (northwestern and northern China) and humid regions (southeastern China) (Fig. 2a). Increasing trends in $PET/P$,
with relatively small magnitudes, occur mainly in the transitional region (northeastern and southwestern China). The spatial
pattern of the $P$ trend is similar to that of the $PET/P$ trend but with the opposite sign, suggesting that the changes in $P$ are
directly linked to changes in $PET/P$ for most of the analysis region (Figs. 2a and 2b). Decreasing trends in $PET$ appear in more





than three-quarters of the analysis domain, but these are significant only in humid regions because of their small magnitudes
(Figs. 2a and 2c).
In the later period, the spatial patterns of the *PET/P*, *P*, and *PET* trends change drastically over the monsoon climate regions
(east of 100°E) (Figs. 2d–2f). The trends in *PET/P* shift from negative to positive values in both the humid (southeastern China)
and arid (northern and northeastern China) regions (Figs. 2a and 2d). These notable alterations of the *PET/P* trend lead to an
increasing trend of overall mean *PET/P* after the early 1980s (Figs. 1a and 2d). Trends in *P* also change significantly: positive
trends are reversed in the arid regions, and the magnitude of the increasing trend decreases in the humid regions (Figs. 2b and
2e). The *P* trends are consistent with the *PET/P* trends in the arid region but not in the humid area (Figs. 2d and 2e). Significant
increases in *PET* explain the inconsistency between the trends in *PET/P* and *P* in the humid area (Figs. 2d and 2f).
The trend shifts that occur around the early 1980s are consistent with regional patterns of changes in climate variables in East
Asian monsoon regions. The variations of *P* are directly associated with the decadal variability of the East Asian monsoon
circulation. As monsoon circulation weakened, both meridional circulation and southerlies decreased over the East Asian
monsoon region; hence, moisture transport is concentrated over southern China (Ding et al., 2008). These changes create
favorable conditions for rainfall over the humid monsoon region but opposite situations over the arid monsoon region. Since
the late 1970s, weakening of monsoon circulation has led to significant decreases and increases in *P* over arid and humid
regions, respectively (Ding et al., 2008; Piao et al. 2010). The increasing trend in *P* over the humid area decreases or reverses
as a result of the reduction in monsoon rainfall related to the recovery of monsoon circulation after the early 1990s (Liu et al.,
2012; Zhu et al., 2012). As a consequence of changes in the monsoon circulation, the decreasing trends in *P* in the arid region
are greater than the increasing trends in the humid area (Fig. 2e). Changes in other climate fields are linked to the positive *PET*
trends (Fig. 2f). For example, the warming trend becomes more severe in the later period (Ge et al., 2013; Yue et al., 2013)
(Figs. S2c and S2g). The trend in absorbed solar radiation changed from dimming to brightening, particularly in the humid
region (Tang et al., 2011) (Figs. S2a and S2e). Consequently, the combined impacts of changes in climate parameters resulted
in the increase in *PET/P*.





### 3.2 Relative influences of five climate parameters on changes in dryness trends

To identify the climate variable that contributed most significantly to the observed *PET/P* trends, we computed the relative

influences of changes in *P*, *Rn*, *WS*, *Ta*, and *RH* on the *PET/P* trends over three hydro-climate regimes (Table S1). Figure 3

displays the averaged effects of five climate parameters and their confidence intervals over the three hydro-climate regimes

for the two analysis periods. Here, positive values of a particular variable indicate increasing rates of *PET/P* with respect to

changes in that variable only, and vice versa. Note that this analysis focuses on the monsoon region, which shows significant

variability in the trends of *PET/P*. Stations located in western China (west of 100°E) are excluded. The mean climate of western

China is distinctly different from the monsoon climate[8]. Furthermore, the dryness trends in these regions are more strongly

associated with variations in *P* for both analysis periods than with other climate variables (Fig. S3).

The relative effects of climate parameters are significantly different according to the analysis period and the hydro-climate

regime, indicating that the mechanisms involved in changing *PET/P* trends operate differently (Fig. 3). Over the arid region,

the positive effects of *P*, *Ta*, and *RH* (1.15%, 0.44%, and 0.55% decade$^{-1}$, respectively) increase the aridity before the early

1980s (Fig. 3a). In addition, the large confidence range of *P* indicates a substantial impact of *P* on the *PET/P* trends locally

(Fig. S3a). In the later period, the change in *P* provides the largest influence (3.27% decade$^{-1}$), at least twice the magnitude of

any other climate parameter. These results imply that the decrease in *P* is the main cause of the significantly increasing trend

in *PET/P* over the arid region. In the transitional region, the negative influence of *Rn* (-0.85% decade$^{-1}$) appears to be the

largest in the earlier period (Fig. 3b), but the wide confidence interval of *P* suggests that *PET/P* trends vary spatially according

to the changes in *P* (Fig. S3a). In the later period, *PET/P* increased because of the positive influences of changes in *P*, *Ta*, and

*RH* (2.02%, 0.97%, and 0.99% decade$^{-1}$, respectively), despite the negative effects of *Rn* and *WS* (-0.34% and -0.48% decade$^{-1}$, respectively). Thus, the increasing trend of *PET/P* in the transitional region is largely a consequence of surface warming

(i.e., *Ta*) and decreases in *P* and *RH*. Over the humid area, negative effects of both *P* and *Rn* (-4.52% and -2.06% decade$^{-1}$,

respectively) lead to the decrease of *PET/P* in the earlier period (Fig. 3c). The contribution from each of the other three

variables is much smaller. In contrast, in the later period, the positive influences of *Ta* and *RH* (0.79% and 1.81% decade$^{-1}$,

respectively) are somewhat larger than the negative influences of *P* and *Rn* (-1.08% and -0.70% decade$^{-1}$, respectively). Thus,



the increasing trend in *PET/P* over the humid region is mainly caused by the warming and subsequent increase in atmospheric
water demand.

## 4 Discussions and Conclusions

The present study suggests that trends in surface dryness reverse from wetting to drying around the early 1980s over both arid
and humid monsoon regions. In addition, major climate parameters determining dryness trends vary by both analysis period
and by region. For the period of 1961-1983, trends in surface dryness are mostly attributed to changes in *P*, regardless of region.
A significant decrease in *Rn* reinforces wetting trends over the humid area by decreasing *PET*. Large influences of *P* and *Rn*
on dryness trends are consistent with the results of previous studies on trends in aridity and *PET* using daily observations of
weather (Wu et al., 2006; Han et al., 2012).
In the later period, changes in *P*, *Ta*, and *RH* lead to drying trends over the monsoon regions. Figure 4 illustrates the impacts
of the three variables on the dryness trend in the arid and humid monsoon regions, respectively. Over the arid monsoon region,
*PET/P* is greatly increased by the positive effects of the three variables, whereas the humid monsoon region shows relatively
small increases in *PET/P* because the positive effects of *Ta* and *RH* are offset by the negative effects of *P*. In contrast to the
importance of the effect of evaporative potential on surface dryness in other water-limited regions (Westerling et al., 2006;
Estes et al., 2012), the decrease in *P* plays a dominant role in the increasing *PET/P* trends in the arid monsoon region. In the
humid monsoon area, the decrease in *RH* shows the largest effect on the *PET/P* trend, despite the relatively small magnitude
of warming. The relationship between air temperature and saturation vapor pressure ($e_s$) (e.g., the Clausius–Clapeyron equation)
explains the large influence of the decrease in *RH*. Due to high mean temperatures in the humid monsoon region (shades of
the map in Fig. 4), warming leads to a steep increase in $e_s$, and a subsequent decrease in *RH*, resulting in a large increase in
evapotranspiration.
Our results based on point observations already include various anthropogenic impacts such as land use/land cover changes
(LULCC) and increased aerosol emissions, which can influence climate and further surface dryness (Menon et al., 2002; Guo
et al., 2013). For example, in the later period, positive influences of *P* are generally inconsistent with negative influences of




$Rn$ (Fig. 3a) because of the decrease in $P$ is favorable condition for the increase in $Rn$, which can result in positive influences
of $Rn$ on the surface dryness trend. We anticipate that aerosols can play an important role in the decrease in $Rn$ in the arid
region by absorbing and scattering solar irradiance. Furthermore, additional heating due to urbanization may cause different
trends in atmospheric water demands between urban and rural areas (Han et al., 2012; Ren and Zhou, 2014). However,
examining the effects of LULCC and aerosols on trends in surface dryness lies beyond scope of the present study.
The effects of $Ta$ and $RH$, which act to dry land surfaces, increased significantly in recent decades in all regions (Fig. 3).
Moreover, over the humid monsoon region, increases in $RH$ show a greater influence on trends in surface dryness than increases
in $P$. This is an unusual situation considering the large variability of summer monsoon rainfall over continental East Asia. The
large influence of $RH$ is supported by steep warming over the humid monsoon area after the early 1980s. This kind of drying
mechanism is consistent with that suggested in assessments dealing with changes in surface dryness during the 20th and 21st
centuries using reconstructed data and future climate projections (Sherwood and Fu, 2014). Thus, our study could be an
observed precursor of the projected drying trends over the humid areas in 21st century (Cook et al., 2014; Yin et al., 2015).
The present results also indicate that drying of the land surface in response to warming is already in progress, not simply a
future risk. Therefore, water management planning must consider the increased water demands associated with warming in
order to mitigate water scarcity, even in the wet monsoon regions.



**Code and data availability**
Codes of NCAR Command Language version 6.3.0, Python, and Interactive Data Language for calculation and climate data
are available upon request to the correspondence author Su-Jong Jeong (waterbell@gmail.com).

**Author Contributions**
C.-E. P. conceived and designed the study, analysed data, and wrote the paper. S.-J. J. helped conceive of the study, and wrote
the paper. C.-H. H. wrote the paper. H. P. analysed data, and wrote the paper. S. P., J. K., and S. F. helped conceive of the
study and wrote the paper.

**Competing interests**
The authors declare no competing financial interest.

**Acknowledgements**
This study was funded by the Korea Ministry of Environment as the "Climate Change Correspondence Program".






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





**Figures**

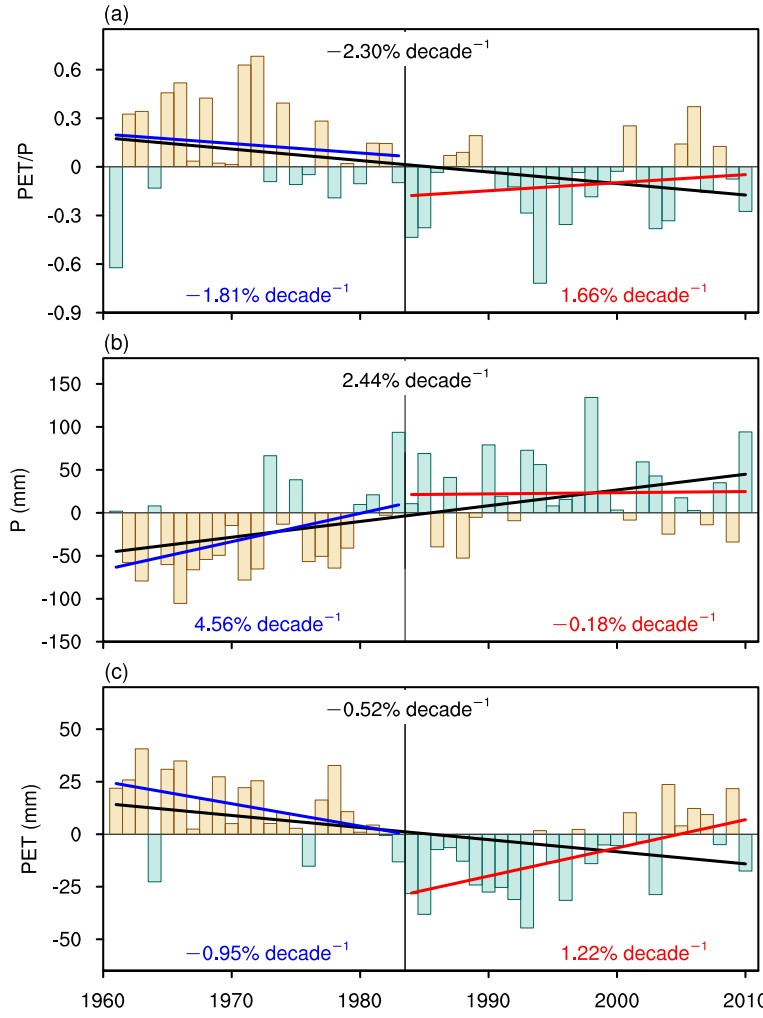


Figure 1: Temporal variations of annual-mean *PET/P* (a), *P* (b), and *PET* (c) in continental East Asia. Yellow and blue bars
indicate the positive and negative anomalies for *PET/P* and *PET*, respectively, but negative and positive anomalies for *P*,
respectively. Black, blue, and red lines are linear regression lines (% decade$^{-1}$) for the periods 1961−2010, 1961−1983, and
1984−2010, respectively.





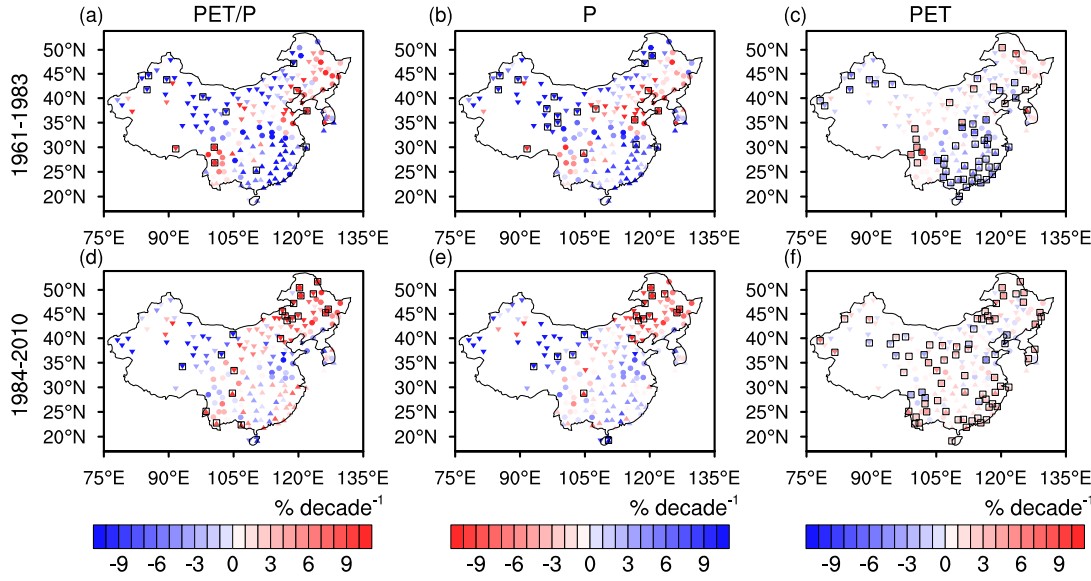


Figure 2: Spatial distributions of trends in *PET/P*, *P*, and *PET* over continental East Asia. a−c: The spatial distribution of trends in annual-mean *PET/P* (a), *P* (b), and *PET* (c) for the period of 1961−1983. d−f: as a−c, but for the period 1984−2010. Inverse triangles, circles, and triangles represent stations classified as arid, transitional, and humid regions, respectively. The empty square indicates that the trend is significant at the 95% confidence level.






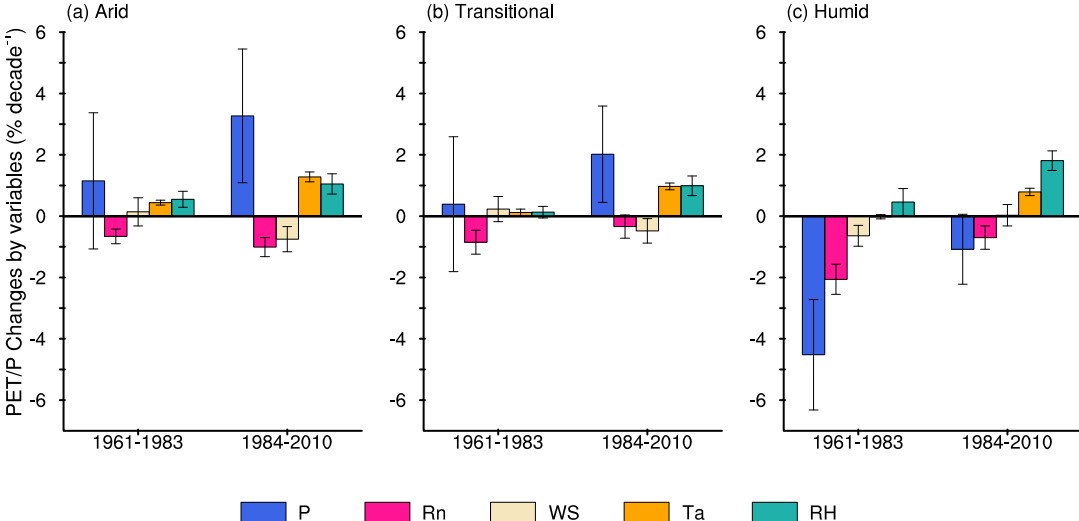


Figure 3: Relative influences (% decade$^{-1}$) of five climate parameters averaged over the three hydro-climate regimes: arid (a),
transitional (b), and humid (c). The influences are computed for the two analysis periods: 1961−1983 and 1984−2010. Blue,
pink, beige, orange, and cyan bars represent the respective influences of $P$, $Rn$, $WS$, $Ta$, and $RH$. Error bars represent confidence
intervals at the 95% confidence level.





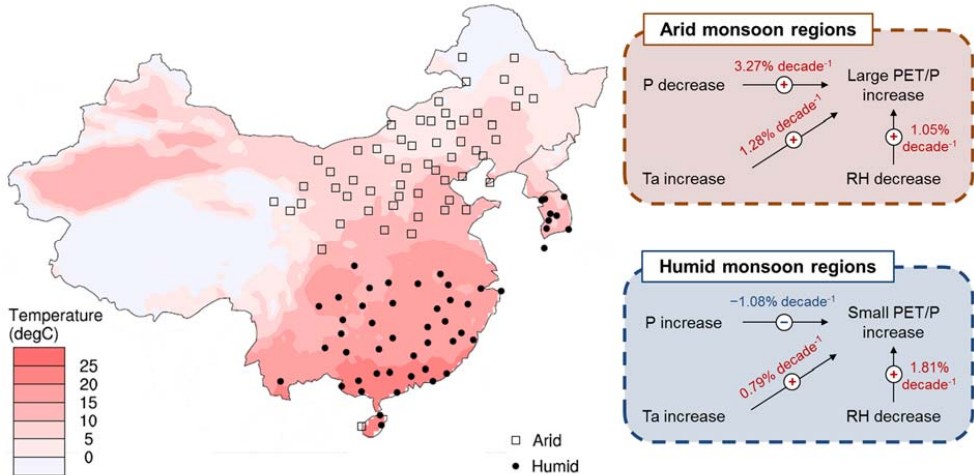


Figure 4: Schematic diagram of the contributions of *P*, *Ta*, and *RH* on the *PET/P* trends in arid and humid monsoon regions
for the period of 1983−2010. Diagrams of the influences of *P*, *Ta*, and *RH* on the trend in *PET/P* over arid and humid monsoon
regions in 1983−2010 are located to the right of annual-mean temperature over continental East Asia for 1961−2010 (°C).
Empty squares and filled circles are stations classified as arid and humid monsoon regions (east of 100°E), respectively.
