# Peer review of "Dominance of climate warming effects on recent drying trends over"

_Atmospheric Chemistry and Physics, 2017_

## Referee Comment (RC1) · Anonymous Referee #1 · 18 Feb 2017

This paper attempts to quantify relative importance of different climate drivers on dryness trends over continental East Asia. The authors find that the drying trends in arid regions are mostly explained by reduced precipitation but it is due to the increase in atmospheric water holding capacity in humid areas. While the topic that aims at understanding influence of different aspects of the climate on dryness is interesting, this paper has number of problems and is not of the quality acceptable for publication. My main concern is related to methods used in the study.

1. It is unclear how the computation is conducted. In particular, how did the author derive the numbers used in Fig. 1? Did they computed the station values first and then average over the region for PET and P separately or did they compute PET/P at individual station and then average over the region? The order of calculation would have an impact on the time series used to plot Fig 1a.

[Figure]

2. It is unclear how the statistical significance of the change point in Fig. 1 was determined. What kind of test for statistical significance was employed for equation (6)? Would the error term epsilon in (5) follow a Gaussian distribution? More importantly, as the authors moving i in (4), the authors are conducting multiple tests. This means that the statistical significance would be incorrect if multiple testing (which the author did not mention) is not explicitly considered. Additionally, Fig. 1 does show long-term trend but the model (5) only considered a step function which is not correct. If a linear trend is considered in (5), would the authors still find a change point around 1980? Note that if there is a long-term trend in the series and if that trend is not considered in the change-point detection, one would always detect a change point in the middle of the time series. This is not useful and it seems that this is what the authors were doing. There is a body of climate literature discussing proper models and tests for the detection of change point but authors do not seem to be aware such studies.

3. The PET calculation (1) involves non-linear interactions among different drivers in particular wind, vapor pressure, and temperature. However, in order to derive the relative importance of different drivers, the authors simplified such interaction by using a linear regression (8). Is such simplification justified? Are the interactions among different drivers too small to be ignored? A proof or references supporting this approach is required. Also, are the regression estimated for individual stations separately or on the regional mean series? These details need to be clearly described for the work to be reproducible. Even if the interaction term among different variables to be small, the variables in (8) may not be independent (e.g., there must be some correlation between radiation and temperature, between temperature and humidity because a day of clear sky would correspond to high radiation, high temperature, and low relative humidity). So how did the authors test the significance of regression?

4. How did the authors estimate the confidence interval in Fig. 3?

5. Fig. 1 does not support the use of step regression of (5). It looks more like a long term trend with the last few years reversed that trend rather than an abrupt change

in the 1980s. This would also invalidate the subsequent analyses regarding different impacts of precipitation and temperature change before and after 1980 as discussed in the paper.

---

## Author Comment (AC1) · 10 Mar 2017

We are deeply appreciate to your critical review and technical comments. Detailed responses to individual specific comments are presented below.

1. It is unclear how the computation is conducted. In particular, how did the author derive the numbers used in Fig. 1? Did they computed the station values first and then average over the region for PET and P separately or did they compute PET/P at individual station and then average over the region? The order of calculation would have an impact on the time series used to plot Fig 1a.

[Reply] Except for directly measured variables (surface air temperature, precipitation, 10m wind speed, sunshine duration, and relative humidity), all variable is computed at each individual station first based on daily observation. After that, we compute annualmean values of variables at individual station, and then compute the mean value for each region to make time series. Thus, the PET/P time series are made by the average of PET/P at each weather site.

2. It is unclear how the statistical significance of the change point in Fig. 1 was determined. What kind of test for statistical significance was employed for equation (6)? Would the error term epsilon in (5) follow a Gaussian distribution? More importantly, as the authors moving i in (4), the authors are conducting multiple tests. This means that the statistical significance would be incorrect if multiple testing (which the author did not mention) is not explicitly considered. Additionally, Fig. 1 does show long-term trend but the model (5) only considered a step function which is not correct. If a linear trend is considered in (5), would the authors still find a change point around 1980? Note that if there is a long-term trend in the series and if that trend is not considered in the change-point detection, one would always detect a change point in the middle of the time series. This is not useful and it seems that this is what the authors were doing. There is a body of climate literature discussing proper models and tests for the detection of change point but authors do not seem to be aware such studies.

5. Fig. 1 does not support the use of step regression of (5). It looks more like a long term trend with the last few years reversed that trend rather than an abrupt change in the 1980s. This would also invalidate the subsequent analyses regarding different impacts of precipitation and temperature change before and after 1980 as discussed in the paper.

[Reply] We think that it is better to response to the second and fifth comment together because of both two comments mention the long-term trend in temporal variation of PET/P. As you commented, there is a significant trend in temporal variations in PET/P ($p > 0.95$) for 1961-2010 shown in figure 1a. This significant trend is because of trends in PET/P are negative at 86.7% of total weather sites (Fig. S1a). However, at most of the individual stations, PET/P trends for 1961-2010 are not significant at a 95% confidence level except the northwestern China (Fig. S1a). A few stations show significant trends over monsoon regions (> 100°E), which we focused on. This spatial distribution of PET/P trends is similar to that of P trends rather than that of PET trends (Fig. S1). The spatial patterns of P trends are well-known results of previous studies: significant increase in P over northwestern China (Zhai et al., 2005; Shi et al., 2007; Piao et al., 2010) and insignificant trends over the monsoon regions (Wang and Ding, 2006; Piao et al., 2010). We will add figure S1 to revised manuscript to show trends in PET/P and P over the monsoon regions are not significant. In addition, we separate the monsoon regions into three regions based on the 50-year climatology of PET/P in the present study: arid (PET/P>2), transient (1<PET/P<2), and humid (PET/P<1) regions. However, the times series of PET/P, P, and PET in figure 1 is based on averages of each variable over whole analysis domain. Figure S2 shows the temporal variations of annual-mean PET/P for 1961-2010 over arid, transient, and humid regions over the monsoon regions, respectively. Magnitudes of PET/P variations are much larger in the arid region than those in other regions. The original time series of PET/P variations may hide variations of PET/P in arid and humid regions. In addition, the linear trend in PET/P variations is not significant in arid and humid regions ($p < 0.9$ for the both regions). The only transient region shows a significant trend in the PET/P variation ($p > 0.95$). Thus, we conclude that the time series shown in figure 1 gives wrong information to readers. We will remove the figure 1 in the revised manuscript.

There are numerous studies about decadal variations in atmospheric circulation and rainfall over the monsoon regions around 1980 (Gong and Ho, 2002; Zhou et al., 2008; Ding et al., 2008; Ha et al. 2012). Based on both insignificant trends over monsoon regions for 1961-2010 shown in figure S1 and background assessments, we can guess easily that there is an abrupt change in temporal changes in PET/P over monsoon regions around 1980. In the present study, we want to verify whether this timing is determined or not based on change-point methods, which can determine undocumented abrupt change. In the original manuscript, we applied two kinds of change-point methods to find a year of abrupt change in PET/P variations shown in figure 1: 1) detection of change-point based on cumulative sum (Pettitt, 1980), 2) detection of change-point

based on simple linear regression model (Elsner et al., 2000). If there is an abrupt change around 1970 or 2000 based on these methods, the change-point is just statistical value and physically meaningless. Fortunately, two methods commonly identify the change-point as 1983, which is generally consistent with decadal variations described in many previous assessments. We concluded that this year, 1983, is a suitable year for separating the period of 1961-2010 into two periods, however, we verify the different time series of PET/P variations over three climate regimes. Now, we apply 3 kinds of change-point methods to temporal variations in PET/P in each region.Details of change-point methods are presented in supplementary.

3. The PET calculation (1) involves non-linear interactions among different drivers in particular wind, vapor pressure, and temperature. However, in order to derive the relative importance of different drivers, the authors simplified such interaction by using a linear regression (8). Is such simplification justified? Are the interactions among different drivers too small to be ignored? A proof or references supporting this approach is required. Also, are the regression estimated for individual stations separately or on the regional mean series? These details need to be clearly described for the work to be reproducible. Even if the interaction term among different variables to be small, the variables in (8) may not be independent (e.g., there must be some correlation between radiation and temperature, between temperature and humidity because a day of clear sky would correspond to high radiation, high temperature, and low relative humidity). So how did the authors test the significance of regression?

[Reply] As you pointed out, equation (8) looks simple considering the non-linear relationship between PET and climate parameters derived in equation (1). However, there are several studies using this linear regression method to determine the most important climate variable for the response of PET to climate changes (Chattopadhyay and Hulme, 1997; Yin et al., 2010; Dinpashoh et al., 2011; Han et al., 2012). We will add this documents to the list of references in the revised manuscript.

To test the significant of regression equation (8), we compute partial correlation coefficients at 189 stations for the period 1961-1983 and 1984-2010 between PET and four parameters, Rn, WS, Ta, and RH (Fig. S4). Regardless of the analysis periods, Rn, WS, and Ta are positively correlated with PET, whereas the partial correlation coefficient of RH is negative. For all four variables, partial correlation coefficients are significant at the 95% confidence level except at few stations, indicating that these fields are closely correlated with PET. Also, significant values of partial correlation coefficients prove that the regression equation does not suffer from multicollinearity of climate parameters. Thus, we can prove the significance of equation (8) and ignore the interaction between climate parameters.

Likewise, the other computed variables, the regression (8) also estimated for individual stations first, and then we compute regional means. We will clarify the order of computation in the revised manuscript.

4. How did the authors estimate the confidence interval in Fig. 3?

[Reply] We calculate the confidence interval at 95% confidence level as the following equation:

$(m-1.96 \times s/\sqrt{n}, m+1.96 \times s/\sqrt{n})$

where, m and s is the mean and standard deviation of relative contributions of each climate variable, respectively. n is the number of stations located in arid (56), transient (50), and humid regions (51), respectively. We will clarify this in the revised manuscript.

Please also note the supplement to this comment:
http://www.atmos-chem-phys-discuss.net/acp-2017-40/acp-2017-40-AC1-supplement.pdf

[Figure]

[Figure]

**Fig. 1.** Spatial distributions of trends in PET/P, P, and PET over continental East Asia. a−c: The spatial distribution of trends in annual-mean PET/P (a), P (b), and PET (c) for the period of 1961−2010.

[Figure]

**Fig. 2.** Temporal variations of annual-mean PET/P over (a) arid, (b) transient, and (c) humid regions where located east of $100°$ E, respectively. Yellow and blue bars indicate the positive and negative values.

[Figure]

**Fig. 3.** The Fc statistics for temporal variations of annual-mean PET/P over (a) arid, (b) transient, and (c) humid regions, respectively.

[Figure]

**Fig. 4.** Spatial distribution of partial correlation coefficient over continental East Asia for 1961-
1983 and 1984-2010 between PET and four parameters such as Rn, WS, Ta, and RH.

**Supplement:**

**Change-point methods**

At first, we try to find change-point of PET/P variations using cumulative sum for three regions following equations (2) and (3) of the original manuscript. The year of abrupt change in PET/P is 1983, 1980, and 1980 in arid, transient, and humid regions, respectively. In the transient region, we apply this method to the detrended time series due to the significant trend, but the result is not changed. In the main body, we omitted the significant test for the results. A simple bootstrap analysis is used to determine a confidence level (Taylor, 2000). Before performing the bootstrap analysis, a difference of the maximum and minimum of cumulative sum as the following equation:

$$C_{diff} = C_{max} - C_{min} \qquad (S1)$$

where $C_{max}$ and $C_{min}$ are the maximum and minimum of cumulative sum. Next, we generate a bootstrap sample of 50 units by randomly reordering values of the original PET/P variations. We compute $C_{diff}^0$ based on the bootstrap sample by performing the same processor following equation (1), (2), and (S1) and determine whether $C_{diff}$ is less than $C_{diff}^0$ or not. If the number of bootstrap sample is $N$, the confidence level of the change-point $\gamma$ is defined as the following equation:

$$\gamma = \frac{x}{N} \qquad (S2)$$

where $x$ is a number of bootstraps which satisfies $C_{diff}^0 < C_{diff}$. We use 5000 bootstrap samples to determine confidence level of the year of abrupt change. The determined the confidence levels are 0.613, 0.996, and 0.954 for arid, transient, and humid regions, respectively. We will add the information about the significant test to the revised manuscript.

The second change-point method is based on linear regression model (Elsner et al., 2000). However, we change the second method to the other method concerning about the possible overestimation of change-point (Lund and Reeves, 2002). The newly adopted method use two simple linear regression model written as the following equation:

$$X_i = \begin{cases} a_1 + b_1 i + e_t, & 1 \le i \le c \\ a_2 + b_2 i + e_t, & c < i \le n \end{cases} \qquad (S3)$$

where $X_i$ is a time series of PET/P variations, $a_1$ and $a_2$ are the intercepts, $b_1$ and $b_2$ are the trend before and after the time of abrupt change $c$. $e_t$ is the error of the linear regression model.

For the time $c$ ($2 \le c \le n - 1$), the parameters of the regression model can be computed based on least squares estimation as the following equations:

$$\hat{b}_1 = \frac{\sum_{i=1}^{c}(i - \bar{\iota_1})(X_i - \overline{X_1})}{\sum_{i=1}^{c}(i - \bar{\iota_1})^2}, and \ \hat{b}_2 = \frac{\sum_{i=c+1}^{n}(i - \bar{\iota_2})(X_i - \overline{X_2})}{\sum_{i=c+1}^{n}(i - \bar{\iota_2})^2} \qquad (S4)$$

$$\hat{a}_1 = \overline{X_1} - \hat{b}_1 \overline{\iota_1}, and \; \hat{a}_1 = \overline{X_2} - \hat{b}_2 \overline{\iota_2} \qquad (S5)$$

where $\overline{X_1}$ and $\overline{X_2}$ are the averages of $X_i$, and $\overline{\iota_1}$ and $\overline{\iota_2}$ are the averages of $i$ before and after time $c$, respectively. The test statistic $F_c$ is represented as the following equation:

$$F_c = \frac{(SSE_R - SSE_F)/2}{SSE_F/(n-4)} \qquad (S6)$$

where

$$SSE_F = \sum_{i=1}^{c} \left( X_i - \hat{a}_1 - \hat{b}_1 i \right)^2 + \sum_{i=c+1}^{n} \left( X_i - \hat{a}_2 - \hat{b}_2 i \right)^2 \qquad (S7)$$

$$SSE_R = \sum_{i=1}^{n} \left( X_i - \hat{a}_R - \hat{b}_R i \right)^2 \qquad (S8)$$

$$\hat{a}_R = 12 \frac{\sum_{i=1}^{n}(X_i - \bar{X})\,i}{n(n+1)(n-1)}, and \; \hat{b}_R = \frac{1}{n} \sum_{i=1}^{n} (X_i - \hat{a}_R i) \qquad (S9).$$

If $c = 1$, the first term in right-hand side of equation (S7) is set to zero. For $c = n$, the second summation of equation (S7) is set to zero. The change-point is determined the time when the maximum value $F_c$ is sufficiently large to exceed the critical values of the $F_{max}$ percentiles (5.91 and 6.92 for 90% and 95% confidence level, respectively; Table 1 in Lund and Reeves, 2002). Figure S3 shows the distribution of the statistic $F_c$ over the arid, transient, and humid regions, respectively. Based on the values of $F_c$, the only transient region shows an abrupt changes in PET/P around 1980. Thus, we can conclude that this method is not suitable to determine abrupt change in PET/P over the monsoon regions.

In addition to the two kinds of change-point methods, we adopt another method, which detects shifts in the mean values between two periods, because of the decadal variations in monsoon circulation and rainfall over the analysis region. This method can be expressed in the form:

$$X_i = \begin{cases} m_1 + e_t, & 1 \leq i \leq c \\ m_2 + e_t, & c < i \leq n \end{cases} \qquad (S10)$$

where $m_1$ and $m_2$ are the means before and after the time $c$ (Beaulieu et al., 2012). As the time $c$ is changed from 1 to $n$, the difference between $m_1$ and $m_2$ ($\Delta m_c$) can be calculated. The abrupt change is determined at time $r$, which satisfies $\Delta m_r = \max(\Delta m_c)$. The year of abrupt change is 1983, 1980, and 1970 over the arid, transient, and humid regions, respectively. The significance test of these years is conducted using student's t-test. The test statistic $T$ is expressed as following:

$$T = \left| \frac{m_{1r} - m_{2r}}{\sqrt{\sigma_{1r}^2/r + \sigma_{2r}^2/(n-r)}} \right| \qquad (S11)$$

where $m_{1r}$ and $m_{2r}$ are the means; $\sigma_{1r}^2$ and $\sigma_{2r}^2$ are the variance before and after the time $p$.

Values of $T$ are 1.870 (p > 90%), 4.744 (p > 99%), and 2.106 (p > 95%) in over the arid, transient, and humid regions, respectively. Considering the significant trend in the transient region, the same analysis is applied to temporal variations in PET/P of the transient region with removing long-term trend. In this change, the change-point is 1980 and the value test statistic is 2.383 (p > 95%).

As we mentioned above, the decadal variation of monsoon circulation around 1980 is a well-known climate shift over monsoon regions. In addition, determined years of abrupt change in PET/P over three climate regimes based on detection methods of undocumented change are generally consistent with the year of climate shift due to decadal variability of monsoon circulation. Thus, we can conclude that separating analysis period into 1961-1983 and 1984-2010 is reasonable to quantify the impacts of climate variables on PEP/P trends. Of course, we approve that descriptions about the methods and background study about the decadal variation of East Asian monsoon circulation are not enough in the original manuscript. We will add sufficient explanations in the revised manuscript.

**References**

Beaulieu, C., Chen, J., and Sarmiento, J. L.: Change-point analysis as a tool to detect abrupt climate variations, Phil. Trans. R. Soc. A., 370, 1228-1249, doi:10.1098/rsta.2011.0383, 2012.

Chattopadhyay, N., Hulme, M.: Evaporation and potential evapotranspiration in India under conditions of recent and future climate change, Agric. Forest Meteorol., 87, 55-73, 1997.

Ding, Y., Wang Z., and Sun Y.: Inter-decadal variation of the summer precipitation in East China and its association with decreasing Asian summer monsoon. Part I: Observed evidences, Int. J. Climatol., 28, 1139–1161, doi:10.1002/joc.1615 2008.

Dinpashoh, Y., Jhajharia, D., Fakheri-Fard, A., Singh, V. P., and Kahya, E.: Trends in reference crop evapotranspiration over Iran, J. Hydrology, 399, 422-423, doi:10.1016/j.jhydrol.2011.01.021, 2011.

Elsner, J. B., Jagger, T., and Niu, X.-F.: Changes in the rates of North Atlantic major hurricane activity during the 20th century, Geophys. Res. Lett., 27, 1743-1746,    doi:10.1029/2000GL011453, 2000.

Gong, D.-Y. and Ho, C.-H.: Shift in the summer rainfall over the Yangtze River valley in the late 1970s, Geophys. Res. Lett., 29, 78-1, doi:10.1029/2001GL014523, 2002.

Ha, K.-J., Heo, K.-Y., Lee, S.-S., Yun, K.-S., and Jhun, J.-G.: Variability in the East Asian Monsoon: a review, Met. Apps., 19, 200-215, doi:10.1002/met.1320, 2012.

Han, S., Xu, D., and Wang, S.: Decreasing potential evaporation trends in China from 1956 to 2005:

Accelerated in regions with significant agricultural influence?, Agric. Forest Meteorol., 154-155, 44–56, doi:10.1016/j.agrformet.2011.10.009, 2012.

Lund, R., Reeves, J.: Detection of undocumented changepoints: A revision of the two-phase regression model, J. Clim., 2547-2554, 2002.

Pettitt, A. N.: A simple cumulative sum type statistic for the change-point problem with zero-one observation, Biometrika, 67, 1, 79–84, 1980.

Piao, S., Ciais, P., Huang, Y., Shen, Z., Peng, S., Li, J., Zhou, L., Liu, H., Ma, Y., Ding, Y., Friedlingstein, P., Liu, C., Tan, K., Yu, Y., Zhang, T., and Fang, J.: The impacts of climate change on water resources and agriculture in China, Nature, 467, 43–51, doi:10.1038/nature09364, 2010.

Shi, Y., Shen, Y., Kang, E., Li, D., Ding, Y., Zhang, G., and Hu, R.: Recent and Future Climate Change in Northwest China, Climatic Change, 80, 379-393, doi:10.1007/s10584-006-9121-7, 2007.

Taylor, W. A.: Change-Point Analysis: A Powerful New Tool For Detecting Changes, WEB: http://www.variation.com/cpa/tech/changepoint.html, 2000.

Wang, B. and Ding, Q.: Changes in global monsoon precipitation over the past 56 years, Geophys. Res. Lett., 33, L06711, doi:10.1029/2005GL025347, 2006.

Yin, Y., Wu, S., Chen, G., and Dai, E.: Attribution analyses of potential evapotranspiration changes in China since the 1960s, Theor. Appl. Climatol., 101, 19-28, doi:10.1007/s00704-009-0197-7, 2010.

Zhai, P. M., Zhang, X. B., Wan, H., and Pan, X. H.: Trends in total precipitation and frequency of daily precipitation extremes over China, J. Clim. 18, 1096–1108, doi:10.1175/JCLI-3318.1, 2005.

Zhou, T., Zhang, L., and Li, H.: Changes in global land monsoon area and total rainfall accumulation over the last half century, Geophys. Res. Lett., 35, L16707, doi:10.1029/2008GL034881, 2008.

---

## Referee Comment (RC2) · Anonymous Referee #2 · 10 Jul 2017

1. Authors needs to bring sense of using their study at regional scale where opinion 'dry gets drier, wet gets wetter' does not fit. It can not be a generalised statement as it is proved over some other regions. 2. How about role of precipitation on humid region- is it only evapotranspiration which is controlling? 3.'Our results suggest that enhanced atmospheric water demands caused by warming can threaten water 28 resources in wet monsoon areas and possibly in other warm and water-sufficient regions' - This process is well understood based on physical laws- then why authors want to claim it that way. 4. All set of equations are from published work and hence need not to part of the main text and can go in the supplementary material. If so, then methodology needs to be simpler for better understanding of common researcher. Overall this work though using important data, but looks more of reporting the finding over the region of

study and lacks in providing comprehension on the physical processes leading to such changes. I am sorry that I can't recommend this paper.

---

## Author Comment (AC2) · 18 Jul 2017

1. Authors needs to bring sense of using their study at regional scale where opinion 'dry gets drier, wet gets wetter' does not fit. It cannot be a generalised statement as it is proved over some other regions.

[Answer] Thank you for your comments. We will modify the sentence and clarify the target region of this study by mentioning the inconsistency of the paradigm over the analysis region.

2. How about role of precipitation on humid region is it only evapotranspiration which is controlling?

[Answer] In humid regions, precipitation change always acts to decrease dryness in

both analysis periods (1961-1983 and 1984-2010). We already described the role of precipitation on dryness trends over the humid area at section 3.2. In the early period, the influence of precipitation is much larger than that of other climate variables, whereas, precipitation is secondly important variable for dryness change in the later period. Then, we wanted to highlight the importance of evapotranspiration over the humid regime in the later period.

3. 'Our results suggest that enhanced atmospheric water demands caused by warming can threaten water resources in wet monsoon areas and possibly in other warm and water-sufficient regions' - This process is well understood based on physical laws- then why authors want to claim it that way.

[Answer] As you mentioned, warming-induced atmospheric water demand increases are the well-known process. However, we want to signify the influence of warming on long-term changes in dryness over wet monsoon area, not the physical law. "Over the monsoon regions, dryness increase due to warming is out of interest in previous studies on dryness trends due to large variation of precipitation". Our results first emphasize the importance of the increase in water demand due to warming for dryness trends over monsoon regions based on site observations, especially in humid areas. We will change this sentence for clarifying the conclusion.

4. All set of equations are from published work and hence need not to part of the main text and can go in the supplementary material. If so, then methodology needs to be simpler for better understanding of common researcher.

[Answer] Thank you for your suggestions, we will move the equation set to supplementary like Han et al. (2012) and Fu and Feng (2014).

---

## Author Response (AR1)

**Point-by-point responses to Referee #1**

This paper attempts to quantify relative importance of different climate drivers on dryness trends over continental East Asia. The authors find that the drying trends in arid regions are mostly explained by reduced precipitation but it is due to the increase in atmospheric water holding capacity in humid areas. While the topic that aims at understanding influence of different aspects of the climate on dryness is interesting, this paper has number of problems and is not of the quality acceptable for publication. My main concern is related to methods used in the study.

[Reply] We appreciate your critical reviews and technical comments. Detailed responses to individual specific comments are presented below.

1. It is unclear how the computation is conducted. In particular, how did the author derive the numbers used in Fig. 1? Did they computed the station values first and then average over the region for PET and P separately or did they compute PET/P at individual station and then average over the region? The order of calculation would have an impact on the time series used to plot Fig 1a.

[Reply] These variables are calculated at each station, then averaged over a region to construct regional means. Except for directly measured variables (surface air temperature, precipitation, 10m wind speed, sunshine duration, and relative humidity), all variables are computed at each station based on daily observation. The annual-mean values, climatologies, and corresponding anomalies are calculated at each station using the daily values. The station values are averaged over each hydro-climate regimes to compute regional means. We have added descriptions about the order of computation in the revised manuscript.

Lines 81-82: We compute daily *PET* and *PET/P*, and then estimated annual-mean values at individual weather sites.

Lines 88-89: To identify the climate variable that contributed most significantly to the observed *PET/P* trends, relative influences of changes in *P*, *Rn*, *WS*, *Ta*, and *RH* on the *PET/P* trends are computed at individual weather sites.

Lines 115-116: Note that the temporal variations are average of *PET/P* anomalies at 56, 50, and 51 weather sites located in the regions of arid, transitional, and humid climate regimes.

2. It is unclear how the statistical significance of the change point in Fig. 1 was determined. What kind of test for statistical significance was employed for equation (6)? Would the error term epsilon in (5) follow a Gaussian distribution? More importantly, as the authors moving i in (4), the authors are

conducting multiple tests. This means that the statistical significance would be incorrect if multiple testing (which the author did not mention) is not explicitly considered. Additionally, Fig. 1 does show long-term trend but the model (5) only considered a step function which is not correct. If a linear trend is considered in (5), would the authors still find a change point around 1980? Note that if there is a long-term trend in the series and if that trend is not considered in the change-point detection, one would always detect a change point in the middle of the time series. This is not useful and it seems that this is what the authors were doing. There is a body of climate literature discussing proper models and tests for the detection of change point but authors do not seem to be aware such studies.

5. Fig. 1 does not support the use of step regression of (5). It looks more like a long term trend with the last few years reversed that trend rather than an abrupt change in the 1980s. This would also invalidate the subsequent analyses regarding different impacts of precipitation and temperature change before and after 1980 as discussed in the paper.

[Reply] Our answers below apply to both the second and fifth comments because these two comments are related to the long-term trend in temporal variation of *PET/P*.

As commented, there is a significant trend in temporal variations in *PET/P* ($p < 0.05$) for 1961-2010 shown in figure 1a of the original manuscript because the trends in *PET/P* are negative at 86.7% of the weather sites examined in this study (Fig. S1a). However, for most stations, *PET/P* trends for 1961-2010 are below the 95% confidence level except for some stations in the northwestern China (Fig. S1a). A few stations show significant trends over the monsoon region (> 100ºE) for which we focus on. This spatial distribution of the *PET/P* trends is more similar the *P* trends than the *PET* trends (Fig. S1). The spatial patterns of the *P* trends are well-known in previous studies: significant increases in *P* over the northwestern China (Zhai et al., 2005; Shi et al., 2007; Piao et al., 2010) and insignificant trends over the monsoon region (Wang and Ding, 2006; Piao et al., 2010). Figure S1 is added to the revised manuscript as figure 3 in order to show that the trends in *PET/P* and *P* over the monsoon regions are not significant for the 1961-2010.

In this study, we separate the monsoon region into three regions based on the 50-year climatology of *PET/P*: arid (*PET/P*>2), transitional (1<*PET/P*<2), and humid (*PET/P*<1). Figure S2 shows the temporal anomalies of annual-mean *PET/P* for 1961-2010 over the three regions in the monsoon region. Because the temporal variations of *PET/P* are much larger in the arid region, figure 1 in the original manuscript may not well show the variations of *PET/P* in the transitional and humid regions. In addition, the linear trend in *PET/P* variations is not significant in the arid and humid regions ($p > 0.1$ for the both regions). Only the transitional region shows a significant trend in the PET/P variation ($p < 0.05$). Thus, we conclude that the time series shown in original figure 1 gives incomplete (and can be misleading) information to readers. We removed the original figure 1 and added figure S2 to the revised manuscript

as figure 4.

There are numerous studies about decadal variations in the atmospheric circulation and rainfall over the monsoon region around 1980 (Gong and Ho, 2002; Zhou et al., 2008; Ding et al., 2008; Ha et al., 2012). Based on both insignificant trends in *PET/P* over the monsoon region for 1961-2010 (figure S1) and the background assessments on the decadal variations of monsoon circulation, we can assume that an abrupt change exists in the temporal changes in *PET/P* over the monsoon region around 1980. Thus, the change-point method is used to examine a year reasonable to divide the analysis period into the pre- and post-transition periods. In the revised manuscript, three change-point methods are used to estimate a timing of an abrupt change in the temporal variation of *PET/P* in each climate regime: 1) detection of change-point based on cumulative sum (Pettitt, 1980), 2) detection of change-point based on simple linear regression model (Lund and Reeves, 2002), and 3) detection of shifts in the mean values between two periods (Beaulieu et al., 2012). Also, the statistical significance of change-points is determined.

At first, we try to find the change-point of the *PET/P* variations for the three regions when a cumulative sum for the *PET/P* variations for the $i$th year ($C_i$) is greatest (Pettitt, 1980). The cumulative sum $C_i$ is calculated as follows:

$$C_0 = 0 \qquad \text{(S1)}$$

$$C_i = C_{i-1} + (X_i - \bar{X}) \qquad \text{(S2)}$$

where $X_i$ is the *PET/P* anomaly in year $i$, and $\bar{X}$ is the averaged *PET/P* for the whole analysis period. The year of abrupt change in *PET/P* is 1983, 1980, and 1980 in arid, transitional, and humid regions, respectively. For the transitional region, we apply this method after removing the linear trend, but the result remains the same. A simple bootstrap analysis is used to determine the confidence level (Taylor, 2000). A difference of the maximum and minimum of cumulative sum is computed as the following equation:

$$C_{diff} = C_{max} - C_{min} \qquad \text{(S3)}$$

where $C_{max}$ and $C_{min}$ are the maximum and minimum of cumulative sum. Next, we generate a bootstrap sample of 50 units by randomly reordering values of the original time series. We compute $C_{diff}^0$ based on the bootstrap sample by performing the same processor following equations (S1), (S2), and (S3) to determine whether $C_{diff}$ is less than $C_{diff}^0$ or not. If the number of bootstrap sample is $N$, the confidence level of the change-point $\gamma$ is defined as the following equation:

$$\gamma = \frac{x}{N} \qquad \text{(S4)}$$

where $x$ is a number of bootstraps which satisfies $C_{diff}^0 < C_{diff}$. We use 5000 bootstrap samples to determine the confidence level of the year of abrupt change. The determined confidence levels are 0.613,

0.996, and 0.954 for the arid, transitional, and humid regions, respectively.

The second change-point method is based on the linear regression model (Lund and Reeves, 2002). Previously, we adopt a method used in Elsner et al. (2000), however, this method can overestimate change-points (Lund and Reeves, 2002). The method uses two simple linear regression models written as the following equation:

$$X_i = \begin{cases} a_1 + b_1 i + e_t, & 1 \le i \le c \\ a_2 + b_2 i + e_t, & c < i \le n \end{cases} \quad \text{(S5)}$$

where $X_i$ is time series of the *PET/P* variations, $a_1$ and $a_2$ are intercepts, $b_1$ and $b_2$ are the trends before and after the time of abrupt change $c$. $e_t$ is the error of the linear regression model.

For the time $c$ $(2 \le c \le n-1)$, the parameters of the regression model can be computed based on a least squares estimation as the following equations:

$$\hat{b}_1 = \frac{\sum_{i=1}^{c}(i - \bar{\iota_1})(X_i - \overline{X_1})}{\sum_{i=1}^{c}(i - \bar{\iota_1})^2}, and\ \hat{b}_2 = \frac{\sum_{i=c+1}^{n}(i - \bar{\iota_2})(X_i - \overline{X_2})}{\sum_{i=c+1}^{n}(i - \bar{\iota_2})^2} \quad \text{(S6)}$$

$$\hat{a}_1 = \overline{X_1} - \hat{b}_1\bar{\iota_1}, and\ \hat{a}_1 = \overline{X_2} - \hat{b}_2\bar{\iota_2} \quad \text{(S7)}$$

where $\overline{X_1}$ and $\overline{X_2}$ are the averages of $X_i$, and $\bar{\iota_1}$ and $\bar{\iota_2}$ are the averages of $i$ before and after time $c$, respectively. The test statistic $F_c$ is represented as the following equation:

$$F_c = \frac{(SSE_R - SSE_F)/2}{SSE_F/(n-4)} \quad \text{(S8)}$$

where

$$SSE_F = \sum_{i=1}^{c}(X_i - \hat{a}_1 - \hat{b}_1 i)^2 + \sum_{i=c+1}^{n}(X_i - \hat{a}_2 - \hat{b}_2 i)^2 \quad \text{(S9)}$$

$$SSE_R = \sum_{i=1}^{n}(X_i - \hat{a}_R - \hat{b}_R i)^2 \quad \text{(S10)}$$

$$\hat{a}_R = 12\frac{\sum_{i=1}^{n}(X_i - \bar{X})\ i}{n(n+1)(n-1)}, and\ \hat{b}_R = \frac{1}{n}\sum_{i=1}^{n}(X_i - \hat{a}_R i) \quad \text{(S11).}$$

If $c = 1$, the first term in the right-hand side of Equation (S9) is set to zero; for $c = n$, the second summation of Equation (S9) is set to zero. The time when the maximum value $F_c$ exceeds the critical values of the $F_{max}$ percentiles (5.91 and 6.92 for 90% and 95% confidence level, respectively; Table 1 in Lund and Reeves, 2002) is selected as the change point. Figure S3 shows the distribution of the statistic $F_c$ over the arid, transitional, and humid regions. Based on the $F_c$ values, only the transitional region shows an abrupt change of *PET/P* around 1980. Thus, we can conclude that there is a trend shift around 1980 in the transitional region. No significant shifts in the *PET/P* trends are fount for the arid

and humid regions.

In addition to the two kinds of change-point methods, we used another method which detects shifts in the mean values between two periods to account for the decadal variations in monsoon circulation and rainfall over the analysis region. This method can be expressed as:

$$X_i = \begin{cases} m_1 + e_t, & 1 \leq i \leq c \\ m_2 + e_t, & c < i \leq n \end{cases} \quad \text{(S12)}$$

where $m_1$ and $m_2$ are the means before and after the time $c$ (Beaulieu et al., 2012). For all $c$ from 1 to $n$, the difference between $m_1$ and $m_2$ ($\Delta m_c$) is calculated. The abrupt change is determined at the time $r$ at which $\Delta m_r = \max(\Delta m_c)$. The years of abrupt change based on this method are 1983, 1980, and 1970 over the arid, transitional, and humid regions, respectively. The significance test of these years is conducted using student's t-test. The test statistic $T$ is expressed as following:

$$T = \left| \frac{m_{1r} - m_{2r}}{\sqrt{\sigma_{1r}^2/r + \sigma_{2r}^2/(n-r)}} \right| \quad \text{(S13)}$$

where $m_{1r}$ and $m_{2r}$ are the means; $\sigma_{1r}^2$ and $\sigma_{2r}^2$ are the variance before and after the time $r$. Values of $T$ are 1.870 ($p < 0.1$), 4.744 ($p < 0.01$), and 2.106 ($p < 0.05$) over the arid, transitional, and humid regions, respectively. The same analysis is applied to the temporal variations in the *PET/P* of the transitional region after removing the long-term trend. In this case, the time of abrupt change is 1980 with the $T$ value of 2.383 ($p < 0.05$).

As mentioned above, the decadal variation of the monsoon circulation around 1980 is a well-known climate shift over the monsoon region. In addition, the three detection methods pick up similar years of abrupt change in *PET/P* over the three climate regions that are generally consistent with the year of climate shift due to decadal variability of the monsoon circulation. Thus, we conclude that separating of the whole analysis period into 1961-1983 and 1984-2010 is reasonable for quantifying the impacts of climate variables on *PEP/P* trends.

[Figure]

**Figure S1.** Spatial distributions of the trends in *PET/P*, *P*, and *PET* over continental East Asia. a−c: The spatial distribution of trends in the annual-mean *PET/P* (a), *P* (b), and *PET* (c) for the period of 1961−2010. Inverse triangles, circles, and triangles represent stations classified as arid, transitional, and humid regions, respectively. The open squares indicate that the trend is significant at the 95% confidence level.

[Figure]

Figure S2. Interannual variations of the annual-mean *PET/P* over the (a) arid, (b) transitional, and (c) humid regions located to the east of 100ºE. Yellow and blue bars indicate the positive and negative anomalies for *PET/P*, respectively.

[Figure]

Figure S3. The $F_c$ statistics for the temporal variations of the annual-mean *PET/P* over the (a) arid, (b) transitional, and (c) humid regions.

3. The PET calculation (1) involves non-linear interactions among different drivers in particular wind, vapor pressure, and temperature. However, in order to derive the relative importance of different drivers, the authors simplified such interaction by using a linear regression (8). Is such simplification justified? Are the interactions among different drivers too small to be ignored? A proof or references supporting this approach is required. Also, are the regression estimated for individual stations separately or on the regional mean series? These details need to be clearly described for the work to be reproducible. Even

if the interaction term among different variables to be small, the variables in (8) may not be independent (e.g., there must be some correlation between radiation and temperature, between temperature and humidity because a day of clear sky would correspond to high radiation, high temperature, and low relative humidity). So how did the authors test the significance of regression?

[Reply] Equation (8) in the original manuscript looks too simple considering the nonlinear relationship between $PET$ and climate parameters derived in Equation (1) of the original manuscript. However, there are several studies using this linear regression method to determine the most important climate variable for the response of $PET$ to climate changes (Chattopadhyay and Hulme, 1997; Yin et al., 2010; Dinpashoh et al., 2011; Han et al., 2012). Thus, the linear regression equation can be used to divide the impact of four climate parameters on $PET$ changes.

To test the significance of the regression equation, we computed partial correlation coefficients between $PET$ and the four parameters, $Rn$, $WS$, $Ta$, and $RH$ at 189 stations for the period 1961–1983 and 1984–2010 (Fig. S4). Regardless of the analysis periods, $Rn$, $WS$, and $Ta$ are positively correlated with $PET$, whereas the partial correlation coefficient for $RH$ is negative. For all four variables, partial correlation coefficients are significant at the 95% confidence level for most stations, indicating that these fields are closely correlated with $PET$. Also, the significance of partial correlation coefficients suggest that the regression equation does not suffer from multicollinearity of each climate parameters. This strongly supports the significance of the regression equation and ignore the interaction between climate parameters.

Similar to other computed variables, the regression equation also estimates for each station at first, then relative influences are computed as illustrated in figure 6 in the revised manuscript.

Details about computing relative influences of climate parameters are described in Section 3 in supplementary information. We add references for the regression equation of $PET$. Also, we describe the order for computing regional mean and test of significance and multicollinearity of the regression equation. Please see the relevant section in supplementary information.

[Figure]

Figure S4. Spatial distribution of partial correlation coefficients over continental East Asia for 1961–1983 and 1984–2010 between *PET* and four parameters such as *Rn*, *WS*, *Ta*, and *RH*. Squared markers indicate that the coefficients are significant at 95% significance level.

4. How did the authors estimate the confidence interval in Fig. 3?

[Reply]    The 95% confidence interval is calculated as below:

$$\left(\bar{x} - 1.96\frac{s}{\sqrt{n}}, \qquad \bar{x} + 1.96\frac{s}{\sqrt{n}}\right) \qquad \text{(S14)}$$

where, $\bar{x}$ and $s$ is the mean and standard deviation of relative contributions of each climate variable, respectively. $n$ is the number of stations located in arid (56), transitional (50), and humid regions (51), respectively.

We add the above description about computing the confidence level in section 3 of supplementary information.

**List of relevant changes made in the manuscript following comments**

1. Lines 81-82. We add sentences about computation of daily mean *PET* and *PET/P* as following:

We computed the daily *PET* and *PET/P*, and then estimated the annual-mean values at individual weather sites.

2. Lines 82-85. We mention about three kinds change-point method applied to temporal variation of *PET/P* as following:

Due to the decadal variation of East Asian monsoon circulation (Ding et al., 2008; Ha et al., 2012), the whole analysis period is divided into two sub-periods, 1961-1983 and 1984-2010, by applying three change-point methods to the temporal variations of *PET/P* (Pettitt, 1980; Lund and Reeves, 2002; Beaulieu et al., 2012, see section 2 of supplementary information for details).

3. Lines 88-89. We add sentences about computation of relative influences of each climate parameters on *PET/P* trends as following:

To identify the climate variables that contribute most significantly to the observed *PET/P* trends, relative influences of changes in *P*, *Rn*, *WS*, *Ta*, and *RH* on the *PET/P* trends are computed at individual weather sites.

4. Lines 104-110. We add a new figure illustrating spatial distribution of *PET/P* trends for 1961-2010 (figure 3) and relevant descriptions as following:

The annual-mean *PET/P* is decreased over most of analysis domain (86.7% of total weather stations) during 1961-2010 by both increase in *P* and decrease in *PET* (Fig. 3). Note that the scale of the *P* trends (Fig. 3b) is reversed in order to represent drying and wetting trends as red and blue colors, respectively. The negative trends in *PET/P* are large and significant at 95% significance level (p > 0.95) over the northwestern China (< 100ºE), whereas the eastern part of the analysis domain (> 100ºE), classified by monsoon climate zone, shows small and insignificant trends in *PET/P* (Fig. 3a). The spatial pattern of the trends in *P* is similar to that of *PET/P* with opposite sign (Figs. 3a and 3b). At more than half of the sites, the trends in *PET* is significant, but the magnitude of *PET* trends is small (Fig. 3c).

5. Lines 113-118. We remove the original figure 1, instead, we add a new time series illustrating annual anomalies of *PET/P* over the arid, transitional and humid regimes (figure 4). Descriptions about this figure are following paragraph:

Figure 4 depicts the temporal variation in the mean *PET/P* for the arid, transitional, and humid regimes over monsoon regions (> 100ºE) expressed as annual mean anomalies. Note that the temporal variations are the averages of *PET/P* anomalies at 56, 50, and 51 weather sites located on arid, transitional, and humid climate regimes, respectively. For all three climate regimes, the *PET/P* anomalies show abrupt

*changes in early 1980s (see supplementary for details). Also, the trends in PET/P anomalies are not significant in the arid and humid regimes. Thus, the analysis of PET/P changes over the monsoon regions needs a separation of the analysis period.*

6. Lines 161-162. We mention about the confidence interval of regional averaged relative influences as following:

*The confidence interval is computed at the 95% significance level based on relative influences of five variables at 56, 50, and 51 stations of arid, transitional, and humid climate regimes.*

7. Supplementary section 2. We describe explanations and results of each change-point methods, also significant tests of determined time of abrupt change.

8. Supplementary section 3. We explain how to compute the relative influences of five climate parameters on *PET/P* trends. In this section, we test significance and multicollinearity of the regression equation of *PET*. Also, we describe the calculation of confidence intervals of relative influences in same section.

[revised manuscript text omitted]